# Biomechanical Forces Determine Fibroid Stem Cell Transformation and the Receptivity Status of the Endometrium: A Critical Appraisal

**DOI:** 10.3390/ijms232214201

**Published:** 2022-11-17

**Authors:** Onder Celik, Nilufer Celik, Nur Dokuzeylul Gungor, Sudenaz Celik, Liya Arslan, Andrea Morciano, Andrea Tinelli

**Affiliations:** 1Department of Obstetrics and Gynecology, Private Clinic, Usak 64000, Turkey; 2Department of Biochemistry, Behcet Uz Children’s Hospital, Izmir 35210, Turkey; 3Department of Obstetrics and Gynecology, School of Medicine, Bahcesehir University, Istanbul 34732, Turkey; 4Medical Faculty, Sofia University “St. Kliment Ohridski”, 1407 Sofia, Bulgaria; 5Medical Faculty, Medical University of Sofia, 1431 Sofia, Bulgaria; 6Department of Obstetrics and Gynecology, “Cardinal Panico” General Hospital, 73020 Lecce, Italy; 7Department of Obstetrics and Gynecology and CERICSAL (Centro di RIcerca Clinica SALentino), “Veris Delli Ponti Hospital”, 73020 Lecce, Italy

**Keywords:** uterine fibroid, fibroid stem cells, endometrium, mechanotransduction, micro-mechanical forces, endometrial receptivity

## Abstract

Myometrium cells are an important reproductive niche in which cyclic mechanical forces of a pico-newton range are produced continuously at millisecond and second intervals. Overproduction and/or underproduction of micro-forces, due to point or epigenetic mutation, aberrant methylation, and abnormal response to hypoxia, may lead to the transformation of fibroid stem cells into fibroid-initiating stem cells. Fibroids are tumors with a high modulus of stiffness disturbing the critical homeostasis of the myometrium and they may cause unfavorable and strong mechanical forces. Micro-mechanical forces and soluble-chemical signals play a critical role in transcriptional and translational processes’ maintenance, by regulating communication between the cell nucleus and its organelles. Signals coming from the external environment can stimulate cells in the format of both soluble biochemical signals and mechanical ones. The shape of the cell and the plasma membrane have a significant character in sensing electro-chemical signals, through specialized receptors and generating responses, accordingly. In order for mechanical signals to be perceived by the cell, they must be converted into biological stimuli, through a process called mechanotransduction. Transmission of fibroid-derived mechanical signals to the endometrium and their effects on receptivity modulators are mediated through a pathway known as solid-state signaling. It is not sufficiently clear which type of receptors and mechanical signals impair endometrial receptivity. However, it is known that biomechanical signals reaching the endometrium affect epithelial sodium channels, lysophosphatidic acid receptors or Rho GTPases, leading to conformational changes in endometrial proteins. Translational changes in receptivity modulators may disrupt the selectivity and receptivity functions of the endometrium, resulting in failed implantation or early pregnancy loss. By hypermethylation of the receptivity genes, micro-forces can also negatively affect decidualization and implantation. The purpose of this narrative review is to summarize the state of the art of the biomechanical forces which can determine fibroid stem cell transformation and, thus, affect the receptivity status of the endometrium with regard to fertilization and pregnancy.

## 1. Introduction

In addition to gynecological lesions originating from the endometrium itself, other pathological lesions of reproductive tissues, such as fibroids, endometrioma, adenomyosis and hydrosalpinx, may adversely affect endometrial receptivity [1,2,3]. These lesions can cause subfertility or early pregnancy loss either by mechanical action or by negatively affecting the production of the endometrial cytokines, growth factors, receptivity genes, and immune cells required for successful implantation [1,3]. Subfertility that occurs regardless of mechanical effects can be defined as non-mechanical [3]. Adenomyosis, which causes subfertility by disrupting the gene expression of receptivity modulators homeobox (HOX) and LIF, is a good example of a non-mechanical effect [4,5,6]. Another non-endometrial pathology that can lead to subfertility by disrupting endometrial receptivity is the endometrioma [7,8]. In recent studies, we have shownthat endometriomas disrupt the expression of homeobox genes and nuclear factor-kappa B (NF-kB) [7,8]. Hydrosalpinx is another non-endometrial pathology that negatively affects receptivity. In addition to the toxic effects of alkaline hydrosalpinx fluid on the developing embryo, endometrial expression of LIF, integrin, NF-kB and homeobox genes wassignificantly decreased [9,10]. All these examples are evidence that a pathology outside the endometrium can lead to subfertility by disrupting the expression of endometrial receptivity modulators [1].

Uterine fibroids located in the myometrium are among the most common causes of subfertility due to their mechanical effects [1,2,3]. Fibroid-derived mechanical effect causes subfertility mainly by preventing sperm–oocyte interaction or embryo passage. Depending on the location and size, fibroids can exert their obstructive effects at the level of the fallopian tubes, cervical canal, or endometrial cavity. Fibroids can also negatively affect fertility by causing involuntary archi-myometrial contractions [11]. The location of the fibroids and their relationship to the endometrium determines the manifestation of subfertility. Submucous or intramural fibroid near the cavity may lead to subfertility. They can inhibit the endometrial wound healing process and favorable conditions for decidualization [2,3]. However, it should be noted that a fibroid does not have to be very large and come into contact with the endometrium to affect receptivity. Fibroids that are small and far from the endometrium may adversely affect the endometrial receptivity by mechanotransduction [12,13]. The process of converting mechanical stimuli into a chemical signal through specialized mechanical transducers is called mechanotransduction [3,14]. Little is known about whether fibroid-induced mechanical stress leads to a change in receptivity through mechanotransduction. The mechanical stress created by a small intramural fibroid (not connected to the endometrium) can be converted into biological signals via mechanotransducers, which can reach the endometrium and disrupt its receptive functions [3,14]. This narrative review was planned to summarize the effects of fibroid-induced micro-mechanical forces on endometrial homeostasis, a mechanosensitive tissue. How fibroid-related mechanical signals transform into biological signals and affect the selectivity and receptivity functions of the endometrium will also be explained with examples.

## 2. Results

### 2.1. Shape and Mechano-Adaptive Properties of the Myometrial Cells

The shape and mechano-adaptive properties of myometrial cells are specialized to adapt to their microenvironment. The set of mechanisms by which micro-mechanical forces (1–10 piconewtons) are converted into biochemical stimuli is called mechanotransduction [12,13,14,15]. Myometrium is one of the most important tissues in which intercellular mechanotransduction works flawlessly. Spatial and temporal control of extracellular cues received via mechanical signals is more efficient than soluble-chemical stimuli, so it has a vital role in maintaining the adaptive and functional properties of the cell. The cell’s response to a mechanical signal is faster and clearer than a soluble-chemical signal response. This affirmative action allows the cell to respond to the most vital of thousands of signals and to eliminate unnecessary signals [15,16]. The mechano-adaptive and plasticity properties of myometrial cells enable the uterus to control the menstrual cycle, pregnancy, and postpartum uterine involution. Myometrium tries to adapt with hyperplasia or hypertrophy against the mechanical effects of the developing fetus, placenta, amniotic fluid and delivery, especially cyclical menstrual changes [17]. Myometrial cells, which have the ability to change their morphology and phenotypic functions, also host uterine fibroids, one of the most common lesions of the reproductive tract. During mechanical transduction, fibroid-associated mechanical signals are converted into biological signals and reach the endometrium. Due to the mechanical stress created by the fibroid-related mechanical forces on the myometrium and endometrium, one out of every four fibroid patients present to a physician with one of the complaints of pain, vaginal bleeding, or subfertility [18]. The selectivity and receptivity functions of the endometrium, which is a tissue surrounded by the myometrium, may be impaired in the presence of fibroids. Fibroids may adversely affect the receptive functions of the endometrium through soluble-chemical signals or by conversion of mechanical stress to biochemical stimuli [3]. As in all cells, cell shape is closely related to its phenotype function in myometrial cells. In order for the cell to perform its phenotypic functions regularly, it must maintain its current shape throughout its life. Cell shape basically enables intracellular and intercellular signal transmission by regulating the arrangement of organelles, the adjustment of the distance between organelles and the distribution of receptors along the plasma membrane [19]. More importantly, with the conserved phenotypic shape, some vital molecules remain in the cytoplasm while transcription factors remain in the nuclear localization. Since cell shape determines the subcellular organization, it is also of vital importance in terms of providing RNA and amino acid transport during transcriptional and translational events [20,21]. In order to carry out division, migration, and intracellular functions, myometrial cells must maintain their dynamic shape and undergo morphological and phenotypic changes when necessary. Myometrial cells adapt to their niches and functions by using chemical and mechanical signals in a coordinated way to maintain their morphological and functional integrity. By mechano-chemical integration, myometrial cells coordinate first with the neighboring myometrial cells, then with the endometrium [22]. The mechanosensitivity of the myometrial cell allows it to respond with a functional or morphological change to piconewton-sized micro-forces exerted by neighboring myometrial cells. The integrity of the plasma membrane and the receptors on it, the number of organelles and their location in the cytoplasm, and the genomic and non-genomic effects of DNA help the myometrial cell to constantly take guiding cues from its microenvironment and to make its decision accordingly. Since the accuracy of the decisions, it takes will ensure the maintenance of both morphological and functional integrity, it determines cell survival [15].

### 2.2. Stimulation of Fibroid Stem Cells by Micro-Mechanical Forces

Uterine fibroids (UF) are fibrotic and clonal tumors that are separated from the surrounding myometrial tissue by a pseudocapsule. However, despite the presence of a pseudocapsule, mechanical communication between the fibroid and myometrium or between the fibroid and the endometrium continues [13]. They originate from uterine smooth muscle fibroid stem cells and reach the highest incidence in reproductive age [23,24]. Despite its high incidence in the general population, fibroids are the only etiological cause of infertility in one or two of a hundred women of reproductive age. In clonality studies using glucose-6-phosphate dehydrogenase, it has been reported that fibroids are monoclonal tumors originating from a single myocyte cell [25]. Since each myoma has a monoclonal development process, it is possible to talk about multiple clones in cases where more than one myoma is detected in the same uterus. Multiclonality is an important feature that explains the fact that one fibroid is in the growth phase while another fibroid is in the shrinking phase [26].

Fibroid stem cells with self-renewal and differentiation capacities are critical for myometrial tissue homeostasis. They are severely affected by the myometrial microenvironment surrounding the cells and determine their behavior accordingly [27]. Micro-forces produced by myometrial cells under physiological conditions can lead to the overproduction of mechanical signals, initiating the evolution of fibroid stem cells towards myoma [3]. Fibroid stem cells can generate soluble signals by being affected by biophysical cues, and fibroid formation is induced. The main mechanism that enables the transformation of normal myometrial cells into fibroid stem cells and then into myomas is considered to be point or epigenetic mutation, aberrant methylation, and abnormal response to hypoxia in the mediator complex subunit 12 gene [28,29,30,31]. The effectiveness of mutations varies according to race, ethnicity, environmental and endocrine factors [29]. Mutations in the MED 12 and high mobility AT-hook 2 genes are sufficient for leiomyoma stem-progenitor cell self-renewal [28,29,30]. While the MED 12 mutation is the most common genetic defect with a frequency of 70% [2,3], AT-hook 2 and fumarate hydratase defects are less common [30,31,32,33]. Exposure to endocrine-disrupting chemicals in the presence of this mutation initiates myoma formation as the DNA repair capabilities of fibroid stem cells will be impaired [32,33].

After the formation of the fibroid stem cell, which is the basic condition for the formation of a new myoma core, activation of different pathways is required for the fibroid to grow in size. Fibroid stem-progenitor cells are defective in terms of estrogen and progesterone receptors expression [34,35,36]. However, these cells grow in size in response to estrogen and progesterone even in the absence of the receptor. This growth is probably mediated by receptors on differentiated myometrial and leiomyoma cells [35,36]. Consistent with this, the Wingless-type/b-catenin pathway has been reported to promote fibroid growth by communicating between leiomyoma stem progenitor cells and the surrounding differentiated cells [31,36]. The growth of a fibroid core consisting of differentiated fibroid cells is almost completely estrogen and progesterone dependent. ERα can be activated both by estrogen binding and phosphorylated by mitogen-activated protein kinase [37]. However, estrogen needs the presence of progesterone to ensure adequate proliferation in fibroid cells. Estrogen increases PR expression in myocytes, allowing myoma cells to respond to progesterone [3,28,29]. Both estrogen and progesterone promote fibroid growth by stimulating moderate myocyte cell proliferation and more intense extracellular matrix synthesis. However, the abnormal increase in extracellular matrix production induces a spontaneous reduction in the size of the fibroid [2,3].

### 2.3. Biomechanical Features of Uterine Fibroids

Fibroid-mediated mechanical signals are transformed into biochemical messages and transmitted to the cells of neighboring and distant tissues. If the fibroid is in contact with the endometrium, mechanical stimulation easily reaches the entire endometrium [38]. In fibroids located far from the endometrium, reaching the endometrium of biomechanical signals may be delayed, but cannot be prevented [3,39]. Fibroids of all locations and sizes can produce mechanical signals to reach the endometrium, although their signal strengths and conduction rates are different. The transition of fibroid-derived mechanical signals from the fibroid to the myometrium or from the myometrium to the endometrium is provided by the mutual interaction of mechanoreceptors and soluble signaling pathways. The histomorphology and mechano-morphological features of the adjacent myometrium and the fibroid itself are the main determinants in the formation and transmission of mechanical signals [12,13,40].

Although fibroids are clonal tumors and originate from somatic stem cells within the myometrium, the stiffness of the fibroid and myometrium is quite different [31]. Because of their high plasticity, fibroid stem cells undergo epigenetic mutations and the cell transforms into tumor-initiating stem cells (TICs). Since the mutation or dysregulation of even one of the fibroid stem cells can initiate fibroid core formation, the differentiation potential of the initial TICs may be effective in determining the fibroid modulus and stuffiness [40,41,42]. Genetic defects that initiate tumor formation in fibroid stem cells occur in genes related to the RNA polymerase II enzyme. The stiffness of uterine fibroids in different locations is 2–4 times greater than the adjacent myometrium [12,41,42]. The dense extracellular matrix (ECM) content, consisting of glycosaminoglycans and interstitial collagen, is the main determinant of fibroid stiffness [40]. The increased stiffness not only makes the mechanical signals stronger, but also allows the signal transmission to be more efficient and faster [15,40]. However, the degree of stiffness that cells in the organism have to generate or respond to a mechanical signal is different for each cell. All cells perceive the mechanical properties of their real niches under physiological conditions and form suitable stiffness and modulus. While one cell has the ability to proliferate and generate signals in intense stiffness, another cell creates a mechanical effect at lower stiffness [43,44]. Since different stiffness values are required for ligand–receptor interaction, each cell creates its own unique micro-forces and stiffness [45]. In many cells (except cancer cells), attachment to a solid basal layer and generating mechanical signals is the preferred mode of communication. Similarly, fibroid stem cells tend to proliferate and generate mechanical signals at higher stiffness values [12].

### 2.4. Generation and Transport of Fibroid-Related Biomechanical Signals to the Endometrium

Mechanical signal transmission from the fibroid to the myometrium or from the fibroid to the endometrium is a solid-state signaling process. For a mechanical signal to be converted to a soluble chemical signal, it must pass through a sequence of pathways that include mitogen-activated protein kinases, A-kinase anchoring protein 13 (AKAP13), phosphatidylinositol-3 kinase/Akt, Janus kinase, and nitric oxide. Rho-GTPases are key enzymes of this pathway [12,40]. The ECM content of the fibroid and Rho-GTPases are the main factors determining the efficiency of this pathway. The amount of ECM is the main component responsible for the stiffness of fibroids, and the modulus and stiffness of the fibroid are significantly higher than the adjacent myometrium [12]. On the other hand, dense stiffness and modulus cause decreased sensitivity of fibroid cells to mechanical stimuli [12,40]. While the ECM and stiffness of the fibroid allow signal formation, they also stimulate tumorigenesis and fibroid growth [46]. However, fibroid growth is not just an ECM-dependent process. Leiomyoma cells have higher levels of active RhoA than myometrial cells. Fibroids have difficulty in responding to mechanical stimuli from their microenvironment, as increased stiffness causes a decrease in RhoA activation. RhoA is a mediator molecule belonging to the Rho family of GTPases and plays an important role in the stiffness of fibroids and in the generation of intra-fibroid osmotic signals. RhoA is intensely expressed in fibroids and to a lesser extent in myometrium [12,40]. Differences in modulus, stiffness, and active RhoA levels between fibroid and myometrial cells may be innate or related to the differentiation capacity of the fibroid stem cell. Since RhoA is involved in the reorganization of the actin skeleton, there may be a defect in communication between fibroid cells and myometrium due to differences inRhoA content [3,12,40,41]. In addition to increased stiffness, a dense vascular network and high fluid content suggest that fibroid-related mechanical signal transduction is not solely ECM or RhoA dependent [3,12,46]. Mechanical signals originating from the fibroid can be directed into the fibroid or into the myometrium and endometrium via the transmembrane receptors’ integrin, cadherin, or caveolin [12]. Signals directed into the fibroid may stimulate fibroid growth, while those directed into the endometrium may impair receptivity (Figure 1 and Table 1).

In addition to the ECM and active RhoA content, the fluid load, vascularity, and osmotic forces of the fibroid are involved in signal transduction [40,46]. Although there are pathways that need to be explained, when a fibroid-derived mechanical signal transforms into a biochemical signal and reaches the endometrium, it may adversely affect healthy decidualization and receptivity. Exposure of the endometrial cavity to air, oil, or mechanical stimuli in rodents improves decidualization [47]. Similarly, when performed in women with implantation failure, mechanical endometrial injury increases the expression of receptivity genes [10]. If a micromechanical force due to physiological sub endometrial myometrium contraction reaches the endometrium, it can stimulate the epithelial sodium channel (ENaC), lysophosphatidic acid (LPA) pathway, proinflammatory cytokines or receptivity gene expression [3]. ENaC activation contributes positively to receptivity by stimulating prostaglandin E2 activation [48]. If mechanical signals bind to LPA receptors in decidual cells, Rho GTPase activation is induced and subsequently the actin–myosin complex becomes functional. This whole process causes a positive progress in decidualization [49]. However, there are no clinical or experimental studies on whether fibroid-mediated mechanotransduction has an effect on endometrial ENaC, LPA, or other receptors. The interaction of mechanical forces and endometrial receptors is affected by many parameters such as the ECM content, stiffness, modulus, size and location of the fibroid [12,40]. Due to the low ECM content in the early formation of the fibroid core, small-intensity microforces can positively affect decidualization. In the later stages, the continuity and decidualization of mechanical signals may be impaired due to increased ECM and stiffness (Table 1).

### 2.5. Impact of Fibroid Related Mechanotransduction on Receptivity

The pathways of propagation of fibroid-induced mechanical stress within the fibroid, adjacent myometrium, and endometrium differ both transcriptionally and translationally. Expression of stress response gene ATF3 mRNA in myoma cells is five times less than in neighboring myometrium cells. Interestingly, the 2.9-fold increase in ATF3 protein expression in western blot analysis is important evidence that myoma-related mechanical stress may play a role in the up- and/or downregulation of genes in the endometrium and myometrium [50]. However, no correlation was found between fibroid diameter and gene expression [50]. Even in the presence of a fibroid smaller than 5 cm, it may cause a change in the expression pattern of at least one gene [2,3]. A crucial question is: “How can a fibroid that does not directly contact and compress the endometrium disrupt the expression of receptivity genes?” The answer to this question lies in the formation of the myoma core. A somatic mutation transforms the myometrial cell into a fibroid stem cell and subsequently triggers the formation of the fibroid core [51,52,53]. However, the main reason for the increase in the size of myoma is fibroblast transformation and deposition of excessive extracellular matrix [53,54]. The random orientation of collagen fibrils in fibroids exerts mechanical stress on the adjacent myometrium [55]. Although it has been clearly shown that mechanical stress turns into a biological signal and reaches neighboring cells in cardiomyocytes, this mode of transmission has not been clearly demonstrated in uterine smooth muscle cells. The mechanical stress that develops due to the increase in ECM and collagen in myoma is converted into intracellular chemical signals by integrins and caveolin proteins (Figure 2). Mechanical stress in both the fibroid and adjacent myometrium has now become a biochemical messenger and spreads to the entire endometrium via the archi-myometrium surrounding the cavity [12,45,56].

A-kinase-binding protein 13 (AKAP13) provides the protein kinase necessary for the function of myometrial filaments. RhoA is the target molecule of AKAP13, which is involved in the cell response to micro-forces [3,12]. Since the subcellular distribution of AKAP13 in myoma cells varies according to the myometrium, the increase in the extracellular matrix provides a chondrocyte-like modulus in the myoma. The RhoA-AKAP13 complex regulates the passage of mechanical stimulus to the myometrium and then to the endometrium via mechanotransducers [3,12,57]. By changing the contractility, migration and proliferation properties of myoma and myometrium fibrils, RhoA-AKAP13 may enable the delivery of biochemical signals to the endometrium and change its receptive properties [12,58]. Rho-GTPases is a molecule that has a critical role in providing the necessary energy during mechanotransduction [59]. In order for biological signals to reach the endometrium, which is a different tissue, and affect receptivity, either the endometrial–myometrial interface must be damaged, or the signal must pass through the estrogen receptors to the endometrium [3,12]. In the presence of fibroids, adenomyotic nodules are formed as a result of myometrial muscle fibers extending towards the basal layer of the endometrium [60]. Adenomyotic nodules allow the endometrial–myometrial interface to deteriorate and the endometrium to grow into the myometrium. This process, called self-propagating, may allow biological signals to reach endometrial cells more easily. Increased local estrogen synthesis due to increased P450arom activity in the fibroid may facilitate signal communication by causing increased peristalsis and tissue damage at the endo–myometrial junction. The fact that the inner myometrium is embryologically different from the outer myometrium may contribute to biological signals reaching the endometrium [2,3]. Erythropoietin secreted by fibroids can help myometrial signals reach the endometrium by regulating angiogenesis and apoptosis in the myoma core, causing rapid growth or destruction [61].

The difference in the effects of fibroids of different sizes and locations on endometrial receptivity may be due to the stiffness and modulus of the fibroid. The modulus is a measure of the elastic deformation of a lesion under a force. It can also be expressed as the force required increasing the unit size of the lesion one-fold. Myometrium has an elastic modulus due to its physiological properties. Fibroids, on the other hand, have a cartilage-like modulus due to their collagen and glycosaminoglycan contents. The cartilage-like modulus gives the fibroid five times more stiffness than the myometrium (5 kPa vs. 15 kPa) [12,58]. The difference in modulus between the myometrium and fibroids is sufficient for mechanical signals to turn into biological signals and to direct them to neighboring myometrial cells and endometrium. For this reason, the mechanical effect of a fibroid at an early developmental stage and that is not sufficiently rigid may not be sufficient to generate a biological signal. A fibroid that cannot initially generate mechanical stress can cause mechanical stress over time and depending on the change in size. In the presence of multiple fibroids at different developmental stages, some fibroids may have a mechanical effect, while others may not. The main factor determining the stiffness of fibroid is not the size but the matrix content [3,12]. Therefore, a fibroid that is small in size but dense in ECM may produce a more intense mechanical stimulus than a fibroid of large size but low in ECM. In this way, the mechanical signals of a small but firm fibroid that does not have direct contact with the endometrium can reach the endometrium. Since submucous fibroids or intramural fibroids compressing the endometrium are in direct contact with the endometrium, they may impair receptivity regardless of the stiffness and modulus of the fibroid. On the other hand, a subserosal fibroid or an intramural fibroid that does not compress the cavity must reach sufficient modulus and stiffness to affect receptivity. It may be possible to clarify this issue thanks to studies involving the stiffness and modulus characteristics of myomas in different locations staged according to the FIGO subclassification system [2,3].

## 3. Discussion

It has been known for a long time that fibroids cause subfertility by disrupting sperm–oocyte interaction or endometrial receptivity [3]. Fibroids can also lead to subfertility by causing an increase in endometrial pan-leukocyte density, a decrease in natural killer cell density, involuntary contraction in myocytes, and the release of transforming-growth-factor-beta 3 (TGF-β3). When fibroid-derived TGF-β3 reaches the endometrium, it blocks bone morphogenetic protein 2 receptors, HOXA10 and LIF expression, leading to insufficient decidualization and implantation failure [3]. Signals due to mechanical forces, on the other hand, block decidualization and receptivity steps through different mechanisms [3]. For the past decade, moderate to good quality studies have been conducted [12,15,40] to determine whether fibroid-induced mechanical forces cause subfertility. In the fibroid stem cell niche, microforces of 1 to 10 piconewton sizes are produced continuously at millisecond or second intervals [12,15]. With these mechanical forces, the cell can both adapt better to its environment and survive by maintaining intercellular communication [13,40]. The translation of mechanical forces into biochemical signals via mechanotransducers such as integrins and caveolin is necessary for the maintenance of phenotypic functions [62]. However, sometimes mechanical signals can disrupt normal physiological dynamics indirectly in neighboring cells and adjacent tissues such as the endometrium [60]. Exogenous mechanical forces can affect the transcriptional or translational pathways in a region or all of the endometrium, preventing the conversion of mRNA to protein or causing a different protein synthesis [38,39,63]. Detection of hypermethylation in the CpG21 region of homeobox genes in the presence of fibroids is evidence that mechanical signals cause changes in protein conformation [64]. The fact that both intramural and submucous fibroids cause methylation defects in receptivity genes is a finding that supports the idea that fibroids have a mechanical effect independent of their localization [3,38,39].

A limited number of clinical studies provide important clues that fibroid-induced mechanotransduction affects endometrial receptivity [39,65]. Although the location, size, and number of fibroids are effective in the response of the endometrium, sometimes a small submucous or an intramural fibroid that does not compress the cavity can adversely affect the receptive pathways of the entire endometrium. Submucosal myomas cause a global decrease in HOXA10 and 11 expressions not only in the endometrial tissue over the fibroid but throughout the entire endometrial tissue, which suggests that the endometrium responds globally to a local myoma compression [38,60]. Indeed, local pressure is an important stimulus for a change in the receptive status of the endometrium. However, the endometrium reacts as if there is compression in the endometrial areas outside the area where it comes into contact with myoma. Response takes place globally, as the mechanical stress that occurs after the local compression of the fibroid on the endometrium and adjacent myometrium is converted into biochemical signals via the mechanotransducers and spread to the entire endometrium [14,58]. Makker et al. [65] reported that 18 patients with intramural fibroids had significantly lower endometrial HOXA10 mRNA, E-cadherin mRNA, and protein levels compared to patients without fibroids. In a recent study, we found a statistically insignificant decrease in HOXA10 and 11 mRNA levels in the presence of intramural fibroids that do not distort the endometrial cavity [39]. We found a 12.8-fold increase in HOXA10 mRNA and a 9.0-fold increase in HOXA-11 after myomectomy. The results of these two studies are important evidence that intramural fibroids that do not have direct contact with the endometrium may adversely affect receptivity through mechanotransduction.

## 4. Materials and Methods

A PubMed search was performed from 2000 to 2022 using the following key terms: “fibroid stem cell”, “uterine fibroid”, “leiomyoma”, “mechanotransduction”, “modulus”, “stiffness”, “receptivity genes”, “endometrium”, “fibroid-derived mechanical signal”, and “implantation”. The clinical and experimental research articles, reviews, and meta-analyses reached on the subject were evaluated in terms of inclusion and exclusion criteria. References to selected articles were reviewed to confirm whether the topic was related to fibroids and mechanotransduction. In the first search, article titles were searched and studies unrelated to mechanotransduction were excluded from the study. Articles evaluating “mechanotransduction”, “uterine fibroid”, “receptivity genes”, and “endometrium”, in human or experimental studies, were selected as narrative review. In the second screening, the abstracts of the papers whose titles met the inclusion criteria, were read by the authors. The full texts of the manuscripts with the abstracts suitable for the selection criteria were accessed and subjected to the third screening. A total of 24 articles meeting the selection criteria were selected, and 13 of them were extensively analyzed for the narrative review (Figure 3).

## 5. Conclusions

The process of converting fibroid-derived mechanical stimuli into a biological signal through special mechanotransducers is called mechanotransduction. Although the location, size, and number of fibroids can affect receptivity, sometimes a small intramural fibroid that does not compress the cavity may adversely affect the receptive status of the entire endometrium. The mechanical stress generated by an intramural fibroid can impair endometrial receptivity after being converted into a biological signal.

## Figures and Tables

**Figure 1 ijms-23-14201-f001:**
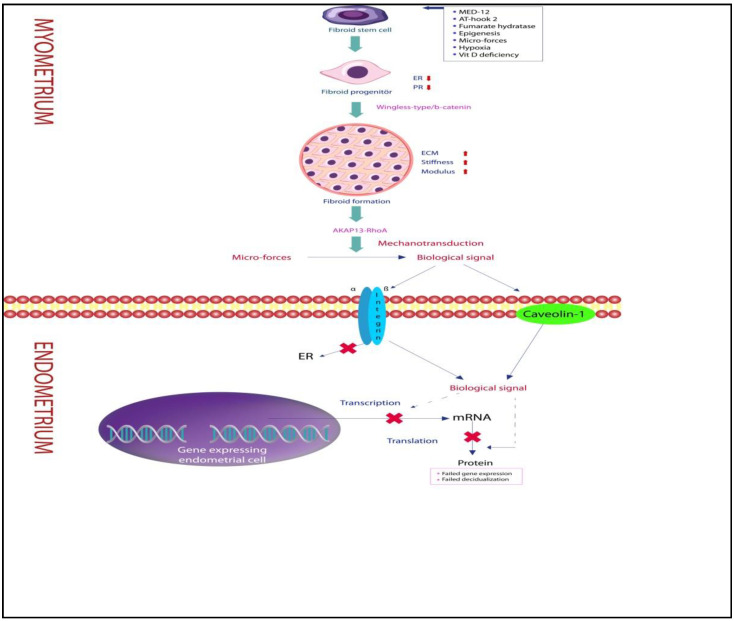
Schematic representation of the stimuli that trigger fibroid stem cell transformation into fibroid progenitor and fibroid formation. Mechanical stimulus transforms into chemical stimulus with solid-state signaling process, reaches the endometrium, and adversely affects receptivity.

**Figure 2 ijms-23-14201-f002:**
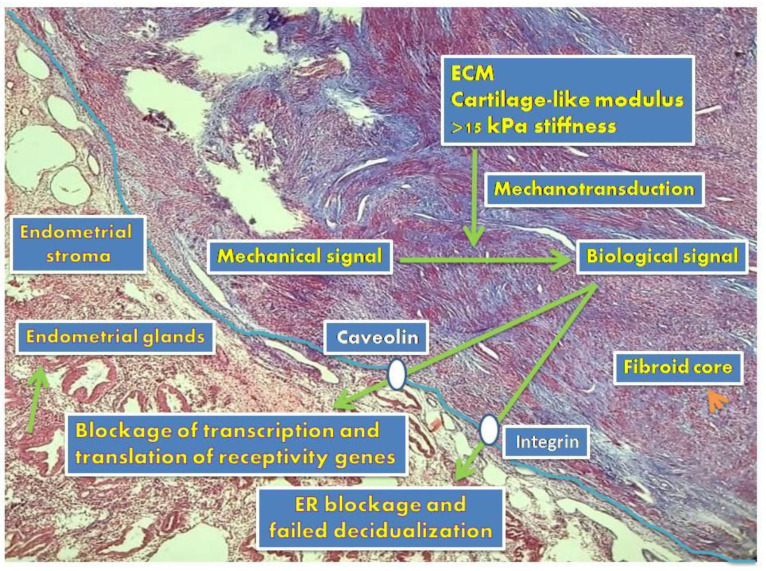
Representative image of Masson trichrome-stained fibroid-endometrium border (blue-green colors show collagen and red shows muscle cells) examined under light microscopy (20×). Fibroid-derived mechanical signals that reach the endometrium by transforming into a biological signal can block the expression of receptivity genes both transcriptionally and translationally. Failed expression of receptivity genes may prevent ovarian sex steroids from acting adequately in the endometrium, leading to insufficient decidualization and implantation defect.

**Figure 3 ijms-23-14201-f003:**
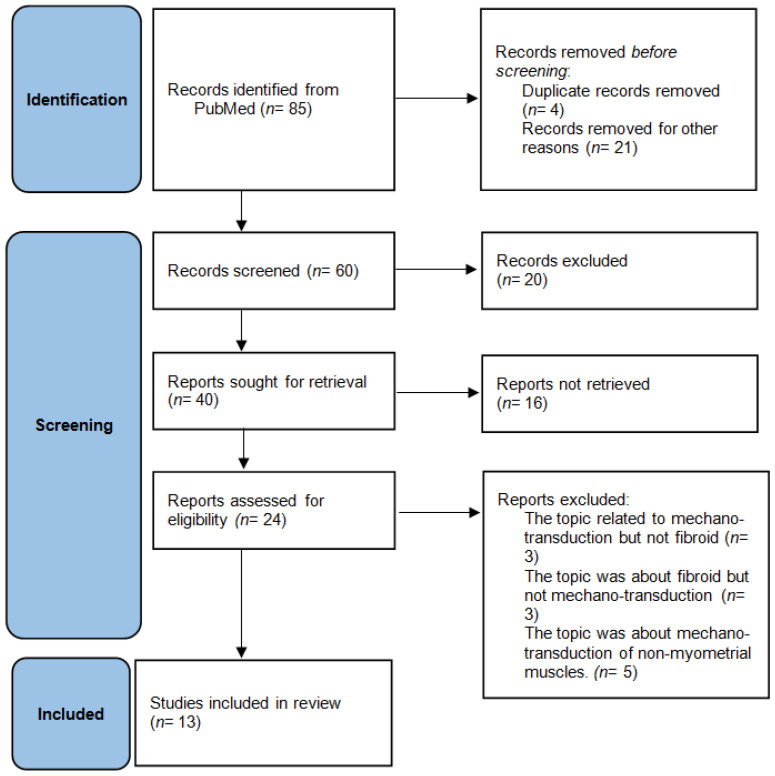
Flowchart used for searching and selecting articles in the narrative review from databases (according to PRISMA 2020).

**Table 1 ijms-23-14201-t001:** Signaling molecules and main mechanisms involved in the transmission of fibroid-derived mechanical signals to the endometrium (Refs. [1,12,13,14,15,16]).

Signal Molecules/Mechanotransducers	Main Mechanisms
ECM	Changes in ECM structure play a role in fibroid formation and growth. The ECM regulates the fibroid cell’s ability to sense and respond to changing microforces. Mechanical signals are transmitted as biological signals from the ECM scaffold via transmembrane receptors to the internal cytoskeleton via integrins, caveolins, and cadherins. Mechanical signals originating from the ECM are converted into biological signals via receptors sensitive to stretching, shear stress, and hydrostatic and osmotic pressures.
Collagen and glycosaminoglycan	Both generate peak stress mediated by the pseudo-dynamic modulus, enabling the mechanical stimulus to be converted into a biological signal stimulus.
AKAP13	AKAP13 is involved in signal generation and transmission through direct communication with the actin cytoskeleton by regulating actin filament nucleation. AKAP13 is extensively expressed in fibroid cells and is involved in the transmission of osmotic signals. RhoA and MAPK are the main targets of AKAP13.
RhoA	RhoA is a molecule belonging to the Rho family of GTPases that switch between non-functional GDP and active GTP. Rho GTPases provide signal transduction by increasing the tension in the scaffold filaments of neighboring cells. Basal levels of active RhoA are equally expressed in myometrial and leiomyoma cells.
AKAP13-RhoA complex	The AKAP13-RhoA complex first stimulates stress-activated kinases followed by actin fibers. Contraction of myofilaments stimulates signal transmission in the endo-myometrial region. The AKAP13-RhoA complex induces the transmission of mechanical signals to the endometrium via estrogen receptors.
Mitogen-activated protein kinases, phosphatidylinositol-3 kinase, Janus kinase, and small GTPases.	They are pathway molecules responsible for possible mechanotransduction between fibroids and endometrium.
Stiffness and Modulus	Myometrium exhibits elastic modulus, whereas fibroids have cartilage-like modulus due to their dense matrix content. The stiffness of the myometrium is about 5 kPa, while the stiffness of the myometrium reaches about 15 kPa. A small fibroid with a large amount of ECM may have a higher stiffness than a large fibroid with a low amount of ECM. Therefore, signals due to an intramural myoma that is not in direct contact with the endometrium but with a stiffness above 15 kPa may spread to the endometrium and impair receptivity.

## Data Availability

Not applicable.

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
