# Peer review of "Biomechanical Forces Determine Fibroid Stem Cell Transformation and the Receptivity Status of the Endometrium: A Critical Appraisal"

_ijms, 2022, doi:10.3390/ijms232214201_

Round 1

Reviewer 1 Report

I appreciate the opportunity to review the manuscript entitled “BIOMECHANICAL FORCES DETERMINE FIBROID STEM CELL TRANSFORMATION AND RECEPTIVITY STATUS OF THE ENDOMETRIUM: A CRITICAL APPRAISAL” submitted to journal International Journal of Molecular Sciences.

The authors conducted the narrative review about biomechanical forces in leiomyoma stem cells, their influence on endometrial receptivity and mechanical transduction, and its biomechanical consequences in uterine leiomyoma with emphasis on clinical consequences such as infertility, subfertility, pregnancy loss, etc.

Reviewer Comments:

1.       Please insert the abbreviations list into the manuscript.

2.       The introduction is too long and not concise. Please rearrange and short introduction, it must have the concise and clear background about the problem review in the article and an aim that must be able to interest readers.

3.       “Most reproductive tract lesions cause subfertility or early pregnancy loss due to their mechanical effects. “   Please specify.

4.       Myometrial cells receive and respond to countless chemical and mechanical signals throughout their lives, day and night, asleep and awake.” Please delete this sentence, it sounds unscientific.

5.       Please insert the table/tables with the main mechanisms and signal molecules involved in mechanical signal transduction.

Taking into account the problems and discussion about biomechanical forces in uterine leiomyoma stem cells as well as its influence on endometrial receptivity my opinion is that this submission meets the criteria to be published in journal International Journal of Molecular Sciences after the minor revisions I suggested.

Author Response

Please insert the abbreviations list into the manuscript.

Comment:  It was provided

AKAP13: A-kinase anchoring protein 13

ECM: Extracellular matrix

TICs: Tumor-initiating stem cells

EnaC: epithelial sodium channel

LPA: Lysophosphatidic acid

TGF-β3: Transforming-growth-factor-beta 3

HOX: Homeobox gene

LIF: Leukemia inhibitory factor

NF-kB: Nuclear factor-kappa B

ER: Estrogen receptor

PR: Progesterone receptor

The introduction is too long and not concise. Please rearrange and short introduction, it must have the concise and clear background about the problem review in the article and an aim that must be able to interest readers.

Comment:  Introduction was shortened. We provided short and clear Introduction.

“Most reproductive tract lesions cause subfertility or early pregnancy loss due to their mechanical effects. “   Please specify.

Comment:  This paragraph has been removed from the text. It was aimed to express those lesions such as fibroid, endometrioma, adenomyosis and hydrosalpinx cause subfertility with mechanical and non-mechanical effects.

“Myometrial cells receive and respond to countless chemical and mechanical signals throughout their lives, day and night, asleep and awake.” Please delete this sentence, it sounds unscientific.

Comment: It was deleted.

Please insert the table/tables with the main mechanisms and signal molecules involved in mechanical signal transduction.

Comment: Table 1 was inserted.

Table 1. Signaling molecules and main mechanisms involved in the transmission of fibroid-derived mechanical signals to the endometrium (Adapted from ref [1,12-16]).

Signal molecules/mechanotransducers

Main mechanisms

ECM

Changes in ECM structure plays a role in fibroid formation and growth. The ECM regulates the fibroid cell’s ability to sense and respond to changing microforces.

Mechanical signals are transmitted as biological signals from the ECM scaffold via transmembrane receptors to the internal cytoskeleton via integrins, caveolins, and cadherins.

Mechanical signals originating from the ECM are converted into biological signals via receptors sensitive to stretching, shear stress, hydrostatic and osmotic pressures.

Collagen and glycosaminoglycan

Both generate peak stress mediated by the pseudo-dynamic modulus, enabling the mechanical stimulus to be converted into a biological signal stimulus.

AKAP13

AKAP13 is involved in signal generation and transmission through direct communication with the actin cytoskeleton by regulating actin filament nucleation.

AKAP13 is extensively expressed in fibroid cells and is involved in the transmission of osmotic signals.

RhoA and MAPK are the main targets of AKAP13.

RhoA

RhoA is a molecule belonging to the Rho family of GTPases that switch between non-functional GDP and active GTP.

Rho GTPases provide signal transduction by increasing the tension in the scaffold filaments of neighboring cells.

Basal levels of active RhoA are equally expressed in myometrial and leiomyoma cells.

AKAP13-RhoA complex

The AKAP13-RhoA complex first stimulates stress-activated kinases followed by actin fibers. Contraction of myofilaments stimulates signal transmission in the endo-myometrial region. The AKAP13-RhoA complex induces the transmission of mechanical signals to the endometrium via estrogen receptors.

Mitogen-activated protein kinases, phosphatidylinositol-3 kinase, Janus kinase, and small GTPases.

They are pathway molecules responsible for possible mechanotransduction between fibroids and endometrium.

Stiffness and Modulus

Myometrium exhibits elastic modulus, whereas fibroids have cartilage-like modulus due to their dense matrix content.

The stiffness of the myometrium is about 5 kPa, while the stiffness of the myometrium reaches about 15 kPa.

A small fibroid with a large amount of ECM may have a higher stiffness than a large fibroid with a low amount of ECM. Therefore, signals due to an intramural myoma that is not in direct contact with the endometrium but with a stiffness above 15 kPa may spread to the endometrium and impair receptivity.

Reviewer 2 Report

Perfect narrative review on mechanotransduction signals potentially leading to subfertility. The purpose of this review manuscript was to summarize the state of the art of the potential biomechanical forces which can determine the fibroid stem cell transformation and affect receptivity status of the endometrium leading to subfertility. The paper is well organized, materials and methods section presents the principles of Pubmed search according to Prisma guidelines. There are no doubts that this article should be published in IJMS. No corrections are needed.

Author Response

We thank the reviewer for the comments reported. The manuscript is the result of a work carried out in a precise and widespread manner, with total collaboration between the authors.

Reviewer 3 Report

Very interesting and well-constructed review work.

There is not much information on the effect of fibroids and their mechanical signals on endometrial receptivity. For this reason, this paper reviews how mechanical signal-derived biological signals affect the selectivity and receptivity functions of the endometrium. I think that is relevant in reproductive field and address a specific gap because this work shows that small intramural fibroid that does not compress the cavity may adversely affect the receptive status of the endometrium.There is scarce information about the role of uterine fibroids in endometrial function from the biomechanical point of view.  Since this is a review paper, I think the methodology is adequate.The conclusions are consistent with the evidence and arguments presented and they address the main question posed. The figures that the work presents are very illustrative and appropriate. I imagine they are from the authors themselves. If not, they would have to mention it.

Author Response

Very interesting and well-constructed review work.

There is not much information on the effect of fibroids and their mechanical signals on endometrial receptivity. For this reason, this paper reviews how mechanical signal-derived biological signals affect the selectivity and receptivity functions of the endometrium. I think that is relevant in reproductive field and address a specific gap because this work shows that small intramural fibroid, that does not compress the cavity, may adversely affect the receptive status of the endometrium. There is scarce information about the role of uterine fibroids in endometrial function from the biomechanical point of view. Since this is a review paper, I think the methodology is adequate. The conclusions are consistent with the evidence and arguments presented and they address the main question posed. The figures that the work presents are very illustrative and appropriate. I imagine they are from the authors themselves. If not, they would have to mention it.

Comment: Figure 1 was drawn by us. Masson trichrome painting in Figure 2 was done by us. The drawings on Figure 2 also belong to us, as the main mechanisms and signal molecules involved in mechanical signal transduction.